Catchment land use predicts benthic vegetation in small estuaries

Cook Perran L.M. perran.cook@monash.edu 1
Warry Fiona Y. 2
Reich Paul 2
Mac Nally Ralph 3
Woodland Ryan J. 4
1 Water Studies Center, School of Chemistry, Monash University , Australia
2 Department of Environment, Land, Water and Planning , Melbourne , Victoria , Australia
3 Institute of Applied Ecology, University of Canberra , Canberra , Australian Capital Territory , Australia
4 Chesapeake Bay Biological Laboratory, University of Maryland, Centre for Environmental Science , Solomons , MD , United States of America
Tonkin Jonathan
Electronic publication date: 2018 Feb 14
Publication date: 2018
Volume: 6
Electronic Location ID: e4378
Received 2017 Nov 9; Accepted 2018 Jan 27
Copyright: ©2018 Cook et al.
Copyright year: 2018
Copyright holder: Cook et al.
License: This is an open access article distributed under the terms of the Creative Commons Attribution License, which permits unrestricted use, distribution, reproduction and adaptation in any medium and for any purpose provided that it is properly attributed. For attribution, the original author(s), title, publication source (PeerJ) and either DOI or URL of the article must be cited.
License URL: https://creativecommons.org/licenses/by/4.0/

Keywords: Nutrient, Land use, Seagrass, Nitrogen, Estuary, Macroalgae

Funding: Australian Research Council LP110100040 The Department of Environment, Land, Water and Planning, Parks Victoria and EPA Victoria This work was supported by the Australian Research Council grant LP110100040, The Department of Environment, Land, Water and Planning, Parks Victoria and EPA Victoria. Fiona Warry and Paul Reich are co-authors on the manuscript and their employment at DELWP is declared.

==============================
Many estuaries are becoming increasingly eutrophic from human activities within their catchments. Nutrient loads often are used to assess risk of eutrophication to estuaries, but such data are expensive and time consuming to obtain. We compared the percent of fertilized land within a catchment, dissolved inorganic nitrogen loads, catchment to estuary area ratio and flushing time as predictors of the proportion of macroalgae to total vegetation within 14 estuaries in south-eastern Australia. The percent of fertilized land within the catchment was the best predictor of the proportion of macroalgae within the estuaries studied. There was a transition to a dominance of macroalgae once the proportion of fertilized land in the catchment exceeded 24%, highlighting the sensitivity of estuaries to catchment land use.

Introduction

Estuaries are well recognized for their ecological and economic value, by supporting diverse natural communities, highly productive fisheries, and recreational amenity (McLusky & Elliott, 2004). Being at the terminus of drainage basins, estuaries are impacted by increased nutrient loads delivered to the coastal zone. Intensive monitoring of both catchments and estuaries has clearly and consistently implicated nutrient loads, in particular nitrogen, as the drivers of multiple adverse ecological responses, including initiation of algal blooms, hypoxia and alteration of secondary production (Hauxwell & Valiela, 2004; Conley et al., 2009). Unfortunately, detailed time-series of nutrient-loading data are not readily available for most estuaries around the world, and a widely applicable and pragmatic approach is required to assess ecological risk and guide land use planning and management targets more generally.

Given that much of the change in nutrient loads is related to human land-use intensity, an alternative approach is to assess ecological risk in estuaries from land-use data, which typically are more readily available than nutrient data (Brinson et al., 2013). The effectiveness of land use data at acting as a proxy for risk to estuaries will depend on the level of detail of classification. Ideally land use should be classified into types of agriculture which have a wide range of nutrient emissions. In practice however, this level of detail is rarely available, and more broad classifications such as forest, urban areas and agriculture are more typically available. In addition to land use, it is important to consider other interacting variables such as estuary flushing time and catchment to estuary area ratio. Estuaries with a fast flushing rate are likely to be less impacted by activities in their catchment than ones with a low flushing rate. Similarly, large estuaries receiving inputs from a small catchment are less likely to be impacted than small estuaries receiving inputs from a larger catchment area.

To link land use to estuary ecological status requires an ecological indicator that is responsive to eutrophication and the identification of a plausible link between that indicator and measureable land-use characteristics. This indicator needs to be both sensitive to changes in nutrient loads and preferably easy to measure. Chlorophyll a (as a proxy for phytoplankton biomass) is a widely used measure of eutrophication that is relatively easy to obtain and strongly related to catchment land use (Meeuwig, 1999). Despite this, phytoplankton dynamics are much influenced by local conditions within estuaries and the often pulsed nature of inputs, resulting in great spatial and temporal variability of chlorophyll a measurements (e.g., Cook, Holland & Longmore, 2010). Therefore, spatially and temporally intensive sampling in an estuary is required for representative and reliable quantification of chlorophyll a. Remote sensing may allow the integration of chlorophyll a concentrations both temporally and spatially, but the complex optical properties of coastal waters has hindered this approach, leading to limited success in relating remotely sensed estimates of chlorophyll a to land use (Le et al., 2015).

An alternative to chlorophyll a that is more stable over short times scales is the ratio of macroalgae area to seagrass area (or macroalgae to total vegetation, MA:TV ratio), which has been shown to increase globally with increased nutrient loading as fast growing macroalgae overgrow seagrass (Hauxwell & Valiela, 2004; Woodland et al., 2015). The generality of the MA:TV ratio, which can be effectively monitored using ground-truthed aerial photographs, suggests that this ratio could provide a suitable proxy for relating remotely sensed land-use characteristics to estuarine eutrophication in shallow estuaries that is easier to obtain and more spatially representative than chlorophyll a.

The aim of this study was to investigate whether land use data available within a national Australian database (Stein, Hutchinson & Stein, 2014) could be used to predict the ecological condition of estuaries. Such information can provide land management agencies with a cost effective and rapid approach to assessing ecological risk to estuaries to enable better prioritization of resources for monitoring and restoration. To do this, we combined previously published data on nutrient loading and estuarine responses with land-use characteristics to compare the efficacy of nutrient load estimates, catchment to estuary area ratio, estuary flushing time and land use as predictors of eutrophication as indicated by the ratio of macroalgae to total vegetation (MA:TV ratio) and chlorophyll a within southern Australian estuaries.

Materials and Methods

The macroalgae to total vegetation (MA:TV) ratio and chlorophyll a data for 14 estuaries in the southeastern Australian state of Victoria are from Woodland et al. (2015). Methods describing field collections, data processing and calculations are described in detail there, so we only briefly outline them here. The estuaries were selected to represent a gradient across land use and nutrient loading, and be geographically representative of the Victorian coastline. Estuary selection also included considerations of total area and geomorphology to avoid scaling-effects arising from large-scale differences in hydrological conditions among estuaries. The MA:TV data represent snapshots in time based on aerial photographs taken between January and February 2012 that were validated by underwater video footage. Video data were reviewed in the laboratory and bottom cover at each drop site was assigned to one or more of the following four primary habitat types: seagrass, macroalgae, bare sediment/unvegetated rocky reef, or channel habitat (>2 m depth). Seagrass and macroalgae habitats were further classified as having sparse–medium (<50%) or dense (50–100%) vegetation coverage. In the case of seagrass habitats with conspicuous epiphytic or intermingled macroalgae, the site was assigned to both habitat categories and each category was assigned a density classification. Spatial mapping was carried out in ArcGIS by constructing habitat raster maps (cell size = 2 m2) based on visual reconciliation of site specific habitat classifications and photographic information from composite aerial images. Vegetated habitat areas were weighted by coverage classifications such that map cells assigned sparse or medium coverage were considered to contain 50% vegetation and dense coverage =100% vegetation. For example, a 10 m2 patch of medium seagrass was designated as having 5 m2 of seagrass habitat and 5 m2 of bare sediment. Total areas of each estuary and each coverage weighted habitat class were calculated and exported for further analysis. Seagrass species were primarily composed of Zostera spp. (includes Z. muelleri and Z. nigracaulis) or Ruppia spp. Macroalgal communities included several genera (e.g., Ulva, Enteromorpha, Hypnea, Gracilaria) associated with eutrophication (McGlathery, 2001).

Surface (c. 0.2–0.5m depth) chlorophyll concentrations (µg L−1) were monitored on two successive outgoing tidal cycles on three separate occasions in a subset of n = 8 estuaries. Sampling occurred once for each estuary during the spring (September–October), early summer (November–December) and late summer (January–February) of 2011–2012. Chlorophyll measurements were taken adjacent to the main channel of the estuary with a calibrated Hydrolab water quality sonde (model DSX5). Concentration values were averaged for each occasion (n = 3–16 observations) and across each of the three seasons to yield an integrated mean chlorophyll concentration in the surface waters of each estuary.

For each estuary and upstream river catchment, potential predictors of variation in the MA:TV ratios were obtained from the National Environmental Stream Attributes database (Stein, Hutchinson & Stein (2014); v1.1.5, Geoscience Australia website: http://www.ga.gov.au). These predictors included four summaries of upstream river catchment land use: proportion modified by human development, proportion with population density ≥ 1 person km−2, proportion urbanized, and proportion receiving or generating fertilizers (predominantly residential areas, grazing pasture, horticulture). We included several covariates that might affect the relationships between the responses and predictors: (1) estuary flushing time (days); (2) the measured areal loading rate of dissolved inorganic nitrogen (DIN) to each estuary (tonnes DIN km−2 of estuary yr−1) (Woodland et al. (2015); and (3) the catchment area to estuary area ratio (C:E). Nitrogen loads were measured based on stream flow and nutrient concentrations measured close to the head of the estuary (salinity = 0), and encompassed >90% of the catchment. River flow (ML d−1) from gauging stations and nutrient and concentration data (mg L−1) for each river system over the 13-yr interval from 2000 to 2012 were obtained by downloading archived data from the Department of Environment, Land, Water and Planning Water Measurement Information System website (data.water.vic.gov.au/monitoring.htm) or provided by Melbourne Water (melbournewater.com.au). We focused our analysis on total nitrogen (TN), oxidized dissolved forms of nitrogen (NO3− and NO2−, hereafter simply DIN). River flow was measured daily; whereas, nutrient sampling intervals ranged from approximately biweekly (n = 23) to quarterly (n = 3–4) with an average of n = 12 samples per river system per year (i.e., monthly sampling). Data were assigned to a 01 June–31 May hydrologic year rather than a calendar year to reflect the annual flow–nutrient cycle responsible for primary production dynamics in Victorian estuaries during the austral summer (Cook & Holland, 2012). Annual loads (Mg yr−1) of TN, DIN, TP and TSS were estimated from measured river flow and concentration data using a flow-stratified Kendall Ratio (Kendall, Stuart & Ord, 1983) approach within a Monte-Carlo simulation-based spreadsheet routine (Tan, Fox & Etchells, 2005). This method has previously been shown to give the same results as those independently published by the Victorian EPA (Cook & Holland, 2012).

There were no sewage treatment plant inputs below the gauging station and atmospheric deposition in this region is negligible (<10%) compared to total loads in these small estuaries. The C:E ratio was included to account for small estuaries fed by a large catchment which would inflate areal nutrient loads and estuary flushing time was included to account for the well-known effect of residence time in modulating eutrophication. To place the gradient of catchment land use intensity within the broader context of the literature, we calculated nutrient export rates from upper and lower quartiles of fertilized catchments (corresponding to >85% and <10% fertilization, respectively) by dividing the total load from the catchment by the total land area of the catchments.

Statistical analysis

We first screened predictors for high collinearity. If there were sets of predictors with pair-wise correlations >0.7, we eliminated all predictors bar one. The retained predictor was the one with the lowest sum of correlations with predictors other than those in the inter-correlated set. Land use and riverine DIN concentrations were highly correlated (r = 0.84), so DIN concentration was excluded from the analysis because it is dependent upon land use. We scrutinized the distributions of the retained predictors. Several were extremely right-skewed, so these were log-transformed (designated by † in Table 1). Once the distributions were near normally or near uniformly distributed, we standardized (mean = 0, standard deviation = 1) the predictors to make the ranges of all predictors comparable and to assist in model convergence.

Table 1 Results of Bayesian variable selection and hierarchical partitioning.

Results of Bayesian variable selection and hierarchical partitioning, which show the predictor variables for the macroalgae to total vegetation (MA:TV) ratio, the posterior probability of inclusion predictor Pr(Inc), the regression coefficient (β), the standard deviation of beta SD(β) and the % of the variability independently explained by each variable. Predictors are abbreviated as follows: C:E ratio is the catchment area to estuary area ratio, Tf is the estuary flushing time, Pop_Prop_1 is the proportion of the catchment with a human population >1 km−2, % Modified is the proportion of the catchment modified by human development, % urbanized is the proportion of the catchment urbanized, % Fertilized the proportion of catchment likely to receive fertilizer inputs, and the Areal.DIN.load is the load of inorganic nitrogen to each estuary normalized to surface area.

Predictor Variable	Pr(Inc)*	β	SD(β)	%indep	
†C:E ratio	0.04	0.01	0.03	6	
Tf	0.02	<0.01	0.03	6	
Pop_Prop_1	0.02	<0.01	0.02	8	
% Modified	0.02	<0.01	0.02	5	
†% Urbanized	0.03	<0.01	0.02	13	
†% Fertilized	1.0	0.30	0.04	46	
†Areal.DIN.load	0.03	<0.01	0.02	15	
Notes.

* Values >0.75 are deemed to be statistically important.

† Ln-transformed.

We used two approaches to identify the potentially important predictors and to identify the relative importance of these predictor variables. First, we used Bayesian variable selection using stochastic search (O’Hara & Sillanpää, 2009). This method identifies those predictors that had high posterior probabilities of being included in the best models for explaining variation in the MA:TV ratio. We used the posterior odds ratio framework to assess predictor importance (Kass & Raftery, 1995). A predictor is assigned an uninformative prior for being included in the best model (a predictor is equally likely to be selected as not), which corresponds to a prior odds ratio of 0.5 (included):0.5 (not included) = 1. If the posterior probability of inclusion, after calculations, is (or exceeds) 0.75, then the posterior odds are 0.75 (included):0.25 (not included) = 3. The ratio of the posterior odds to the prior odds is the posterior odds ratio (here 0.75:0.25/0.5:0.5 = 3), with values exceeding 3 being indicative of probable importance of a predictor in explaining variation in the response variable (here MA:TV ratio). Models were calculated using JAGS (Plummer, 2003).

Second, we used hierarchical partitioning (HP) on the predictor variables to calculate the relative proportions independently explained by each predictor. We used the hier.part (Walsh & Mac Nally, 2004) package in R (R Development Core Team, 2011). HP complements Bayesian model selection by quantifying the relative amounts of variation independently explained by each predictor (Mac Nally, 1996).

The %fertilized predictor proved to be important (Table 1) but we were concerned that its effect might be moderated by the catchment to estuary (C:E) ratio or residence time of the estuary (Tf). Therefore, we used Bayesian model selection and HP analyses for a full interaction model involving these three predictors, notwithstanding that the C:E ratio and Tf were not important from the analyses in Table 1. The full interaction model included the three predictors, each pair of interactions, and the three-way interaction (see Table 2).

Table 2 Results of Bayesian variable selection for potential interaction terms.

Results of Bayesian variable selection for potential interaction terms showing the posterior probability of inclusion predictor Pr(Inc), the regression coefficient (β), the standard deviation of beta SD(β) and the % of the variability independently explained by each variable (as for Table 1) computed using hierarchical partitioning.

Interaction terms	Pr(Inc)	β	SD(β)	% indep.	
†C:E ratio	0.067	0.016	0.026	12	
Tf	0.025	−0.004	0.019	6	
†% Fertilized	1.0	0.294	0.042	52	
†C:E ratio × Tf	0.016	0.002	0.016	3	
†C:E ratio ×†% Fertilized	0.068	−0.013	0.034	14	
Tf ×†% Fertilized	0.023	0.006	0.018	4	
Tf ×†C:E ratio ×†% Fertilized	0.030	−0.008	0.019	9	
Notes.

Values >0.75 are deemed statistically important.

† Ln-transformed.

We fitted a change-point model for the relationship between MA:TV and %catchment fertilization (F). The model was: MA:TVi∼Normalμi,σ,

μi=α∗δFi−γ+β1∗δγ−Fi∗Fi+β2∗δFi−γ∗Fi;

where: δ is unity if the argument is non-negative and zero otherwise and γ is the change-point. The priors were: α,βi∼Normal0,σ=2,andγ∼Uniform0,100. The program JAGS (Plummer, 2003), which uses a Gibbs sampler, was used to fit the relationship; there were 12,000 iterations and 5,000 ‘burns-in’ samples. We checked convergence using Gelman–Rubin methods (convergence of multiple independent chains).

Results

The proportion of fertilization within a catchment was the only important predictor of the MA:TV ratio within our set of estuaries (Table 1). There was little evidence that interactions between the proportion of fertilization and residence time or the ratio of the catchment area to estuary area were important (Table 2). Macroalgal dominance in estuaries was positively associated with an increasing proportion of the catchment receiving fertilizers (Fig. 1), with macroalgal cover dominating benthic vegetation in estuaries with catchments that had >24% of the catchment with fertilization. The change point model was a good fit to the data, with R2 = 0.93 (Fig. 1). The change-point γ was estimated to be 24.3 ± 9.9 SD (% catchment fertilization). The intercept, α, was 0.675 for points above the change-point and 0 otherwise. The slopes were: β1 = 0.024 (slope for points below the change-point) and β2 = 0.002 (points above the change-point). There was little evidence for a relationship between chlorophyll a and % catchment fertilization (linear regression, R2 = 0.14, P > 0.05), although all estuaries with catchments >24% fertilized land had average chlorophyll a concentrations >6 µg/L during the late spring to late summer growth period. Similarly, there was little evidence for a relationship between the proportion of the estuary area covered by benthic vegetation and proportion of the catchment receiving fertilizers (R2 = 0.17, P > 0.05). Total nitrogen (N) and dissolved inorganic nitrogen (DIN) exports were 0.22–1.7 and 0.063–0.74 kg ha−1yr−1 respectively for the catchments in the lower quartile of fertilization (<10% area fertilized) and 1.9–6 and 0.7–3.1 kg ha−1yr−1 for the catchments in the upper quartile of fertilization (>80% area fertilized).

Figure 1 Plot of MA:TV vs % of catchment fertilised.

The ratio of macroalgae to total vegetation (MA:TV) versus the % of the catchment receiving fertilizer inputs. Scatterplots show observed (open circles) and fitted (solid circles) for the change-point analysis, with the estimated position of the change-point shown by a dashed vertical line.

Discussion

Land use as a predictor of shallow benthic vegetation

The results show that land use is a strong predictor of the proportion of macroalgae to total vegetation within south-eastern Australian estuaries. Although the current analysis shows that land use is a stronger predictor than nitrogen loads, we do not interpret this to mean that nitrogen inputs to estuaries are not the key driver of changes to estuarine benthic vegetation. Rather we use these findings to shed light on the possible mechanisms through which nutrients drive change within estuaries and how catchment land use integrates this change.

The proportion of the catchment receiving fertilization was a better predictor of the MA:TV ratio than was areal dissolved inorganic nitrogen (DIN) load, which we have previously suggested to be a better predictor of MA:TV in these estuaries than total nitrogen (TN) or total phosphorous (TP) loads (Woodland et al., 2015). This outcome arose because there was a relatively weak relationship between measured loads (both total and normalized to the estuary area) and the total area fertilized (km2) within catchments (R2 = 0.33). The lack of such a relationship is consistent with previous studies that have shown nitrogen attenuation factors can be highly variable (Elwan et al., 2015). Therefore, the relationship between land use and estuarine response is not just driven by a land-use-load relationship, as we had expected.

There was a strong non-linear relationship between DIN concentration in the rivers and the MA:TV ratio (R2 = 0.78, when DIN concentration is log transformed) arising from the relationship between land use and DIN concentrations within the rivers (Woodland et al., 2015). However, we do not believe that the nutrient concentrations observed within the rivers are the primary driver of changes in the MA:TV ratio because these rivers drain into estuaries of greatly different sizes and hence dilution. Moreover, there was no relationship between DIN and the MA:TV ratio when the nutrient concentration was normalized to estuary area. These results suggest that catchment land-use metrics ‘integrate’ factors affecting the amount and availability of nutrients within the estuary that control the MA:TV ratio, which are missed by instantaneous measurements of load. Catchment land-use metrics may incorporate: (1) the historical sequence of delivery of nitrogen (N) and total suspended solids that are trapped and recycled or re-suspended within the estuary; (2) increased bioavailability of particulate and dissolved organic N delivered to estuaries as fertilization increases (Seitzinger, Sanders & Styles, 2002; Petrone, Richards & Grierson, 2009); and (3) local groundwater inputs of N directly to estuaries (Wong et al., 2014).

Our results suggest that estuarine vegetation structure can be substantially altered when agricultural land use constitutes as little as 24% of the catchment. Therefore, it is instructive to compare the nutrient loading and export rates measured here with previous studies to place the land-use intensity in this study in a wider context. The areal loading rates of N in these estuaries span the range reported globally for estuaries, ranging 102–105 mmol N m−2 yr−1 as total N (Woodland et al., 2015). The rates of N generation from the catchments in the lowest quartile of %fertilization averaged 0.9 and 0.25 kg ha−1 yr−1 for TN and DIN respectively (Table 3), which are at the lower end of DIN exports of 0 to 10 kg ha−1 yr−1 reported in forested catchments (Bernal, Butturini & Sabater, 2005; Brookshire et al., 2012). Our undisturbed catchments have lower exports than forested catchments elsewhere in the world, highlighting the relatively oligotrophic state of estuaries fed by pristine catchments in Australia. For the most fertilized catchments (>80% fertilization), we saw average N generation rates of ∼4.5 and 1.9 kg ha−1 yr−1 for TN and DIN respectively, which are comparable with reported nutrient generation rates for mixed farming/rural land use in southeastern Australia (Drewry et al., 2006). Our N generation rates are at the lower end of reported nutrient generation rates of 4–14 kg ha−1 yr−1 for TN in European and North American systems (Howarth et al., 1996), highlighting that even small amounts of relatively low-intensity agriculture can lead to large changes in benthic vegetation in these naturally oligotrophic estuaries. Studies from other locations are needed to investigate whether the patterns observed here are globally applicable.

Table 3 Nutrient export rates for total nitrogen (TN) and NOx for the catchments in this study.

Comparisons for exports from forest and mixed farming are given for SE Australia. % Fertilized exports from catchments are all given in kg ha−1 y−1. Published export rates for Australian forest and mixed farming/rural land uses are from Drewry et al. (2006).

System	%_Fertilized	NOx	TN	
Wingan River	0.49	0.06	1.7	
Cann River	2.0	0.08	0.22	
Genoa River	4.7	0.11	0.62	
Aire River	13	0.74	1.0	
Gellibrand River	25	1.3	3.1	
Merriman Creek	35	0.38	1.1	
Tarra River	38	1.6	2.5	
Werribee River	56	0.11	0.35	
Patterson River	57	0.33	1.2	
Glenelg River	63	0.24	0.65	
Kororoit Creek	82	0.28	0.56	
Tarwin River	85	3.1	6.1	
Curdies River	86	0.70	2.4	
Bass River	92	3.1	7.7	
Moyne River	98	0.79	1.9	
Forest	–	–	0.9–2	
Mixed farming/rural	–	4	0.5–4.5	

The use of the macroalgae to total vegetation ratio as an indicator of estuarine condition

It is virtually impossible to select an ecological indicator that represents all critical aspects of ecosystem function. Our choice of MA:TV was based on the requirement that we could easily obtain relevant data for large areas of the estuary. In shallow estuaries, such as those studied here, macroalgae is widely considered an indicator of eutrophication (Valiela et al., 1997). There are cascading ecological consequences from the increasing dominance of macroalgal biomass to food webs, from changes in consumer biodiversity, productivity and trophic relationships (e.g., omnivory) to biogeochemical cycling and dissolved-oxygen dynamics (Sogard & Able, 1991; Valiela et al., 1997). Consistent with this, we also saw that once catchment fertilization exceeded 24%, and alongside a transition to macroalgal dominance of demersal vegetation, all chlorophyll a measurements were >6 µg/L, which typically is regarded as eutrophic (Hakanson, Bryhn & Hytteborn, 2007).

The ability to reliably predict MA:TV ratio using just one variable differs from previous studies that have shown that multiple predictors are needed to explain >50% of the variation of other response variables (Li et al., 2007; Greene et al., 2015). One of the strongest relationships from previous reports has been between chlorophyll a and the area of agriculture land use in a catchment and estuary volume (R2 of 0.68) in 14 Canadian estuaries (Meeuwig, 1999). This is not unexpected because a certain nutrient load will be diluted to different extents depending on estuary volume. Similarly, one would expect phytoplankton concentration to be sensitive to estuarine residence time, which will lead to different wash-out rates (Nixon et al., 2001). We saw no clear relationship between chlorophyll a and percent fertilization in our data set, which was consistent with the need for other variables to describe the response of this parameter.

We found no important relationship between seagrass or total vegetation areal extent and land use. Elsewhere, a combination of land-use and physical factors, such as tidal range and mean wave height, were needed to describe seagrass areal extent (Li et al., 2007), illustrating the interplay of factors other than eutrophication in controlling seagrass distribution. By standardizing macroalgal extent as a proportion of total vegetation, our analysis reduces the influence of physical factors, such as sediment movement and hydrodynamics, that often limit the growth of benthic vegetation. The MA:TV measure also accounts for estuaries with different hypsometric profiles because the MA:TV ratio is functionally constrained to those areas where light penetration can support benthic vegetation. Change point analysis showed that the MA:TV ratio increased at a lower rate above a catchment fertilization of 24%. This probably suggests that any increase in biomass above this point may have manifested itself as increased thickness (as opposed to area). Alternatively, macroalgae became growth limited at high biomass due to limitation by other factors such as light and/or space. Therefore, a disadvantage of this approach is that it does not sensitively distinguish between moderate and high levels of disturbance.

The interaction model showed that estuary flushing time did not contribute much to explaining variation in the MA:TV response. The residence times used in our study are relatively short (0.6–4.2 days compared to months to years for lagoons), which may partially explain the lack of importance of residence time. However, macroalgae can assimilate N in several hours and, given the subsequent relatively slow turnover of N, residence time may not significantly affect the macroalgal response (Nixon et al., 2001). Lagoon systems, with much longer residence times, are likely to respond differently to our estuaries because phytoplankton are more dominant in systems where water residence time exceeds phytoplankton turnover time (Hauxwell & Valiela, 2004). As the catchment-estuary area ratio (C:E ratio) increased, we expected that nutrient inputs would be distributed over a smaller estuarine area and may render estuaries more sensitive to the proportion of fertilizing land uses in the catchment. The results of the interaction model, which included C:E ratio (Table 2), suggested that there was little evidence of an interaction of the C:E ratio with the proportion of fertilizing land uses in the catchment. As the C:E ratio increased, the transit time of loads delivered to the estuary decreased, leading to lower retention and exposure to nutrients within the system compared to estuaries with lower C:E ratios. Systems with low C:E ratios have the load spread over a larger area, but with a longer transit time, leading to higher retention and exposure to nutrients. Our results suggest that these opposing effects may largely cancel each other out, leading to the C:E ratio having no perceptible effect on MA:TV.

Application to management

Although nutrient loads are a critical management tool for receiving waters, the expense and long time frame required to collect and meaningfully interpret these data mean that such data are not always available. However, land-use information typically is much more readily available, and as has been illustrated here and previously (Meeuwig, 1999; Meeuwig, Kauppila & Pitkänen, 2000), can provide a good indicator of likely risk to estuaries. Once estuaries at risk have been identified, there should be a further assessment of ecological impact. The MA:TV ratio provides a relatively rapid and spatially representative indication of ecological response and condition. By way of an example case study, an Index of Estuary Condition (IEC) is being used to assist the prioritization of estuary management investment thereby supporting the Victorian Waterway Management Program, Australia (DEPI, 2013). Indicators that have demonstrated relationships with processes threatening estuaries (e.g., nutrient loading and land-use change) are essential if broad scale resource condition assessments are to be interpretable, ecologically meaningful and useful for management (Barbour et al., 2000; Stoddard et al., 2008). For these reasons the MA:TV index is being incorporated into the Victorian IEC.

Conclusion

The proportion of catchment fertilization is a strong predictor of the proportion of macroalgae relative to seagrass in small south-eastern Australian estuaries. Our results suggest that estuaries are sensitive to land-use change, and that conversion of as little as 24% of a catchment to fertilized land uses can substantially shift the dominance of benthic primary produces from seagrass to macroalgae. The use of simple land-use measures may provide a strong indicator of risk of estuarine eutrophication where other data are absent. Further studies across a wider geographic and climatic spread are required to investigate relationships between catchment land use and estuarine vegetation globally.

Supplemental Information

Supplemental Information 1 Raw data

Raw data used in the analysis.

Click here for additional data file.

We thank Walter Boynton and Francis Burdon for helpful suggestions on the manuscript.

Additional Information and Declarations

Competing Interests

Author Contributions

Data Availability

Perran Cook is an Academic Editor for PeerJ. Fiona Warry and Paul Reich work for the Department of Environment, Land, Water and Planning, Victoria, Australia.

Perran L.M. Cook conceived and designed the experiments, performed the experiments, analyzed the data, contributed reagents/materials/analysis tools, wrote the paper, prepared figures and/or tables, reviewed drafts of the paper.

Fiona Y. Warry and Paul Reich conceived and designed the experiments, wrote the paper, reviewed drafts of the paper.

Ralph Mac Nally analyzed the data, contributed reagents/materials/analysis tools, wrote the paper, prepared figures and/or tables, reviewed drafts of the paper.

Ryan J. Woodland conceived and designed the experiments, performed the experiments, analyzed the data, contributed reagents/materials/analysis tools, wrote the paper, reviewed drafts of the paper.

The following information was supplied regarding data availability:

The raw data is included as a Supplemental File.

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
