# Peer review of "Catchment land use predicts benthic vegetation in small estuaries"

_PeerJ, doi:10.7717/peerj.4378_

## Round 0.1 · original submission · Major Revisions

Many thanks for your contribution.

We have received two generally positive reviews, which provide several important suggestions to improve the manuscript. Reviewer 2 highlights some useful suggestions for potential statistical analyses that may help to uncover more from the data. Given the clear non-linear threshold response, and the emphasis on 20% fertilised land shifting estuarine vegetation into a new state, I think Reviewer 2 is justified in requesting some more formal means of examining the threshold, such as changepoint analysis. I agree with both reviewers that considerably more information is needed on the statistical approach, the model details, the interactions, the source of R2 and p values, and the specific functions used. For instance, expand on the use of uninformative priors, rather than simply stating "Given uninformative priors for predictor inclusion..." This additional required detail also applies to the results. Results of variable selection are given but not of overall model results. Reviewer 1 asks where the R2 values come from. The methods state linear regression, but there are clearly non-linear responses. At L148-148, you state "a strong non-linear relationship between DIN concentration in the rivers and the MA:TV ratio (R2 = 0.78..." but according to your methods, you ran linear regression. What do the R2 values represent then?

In addition to the points raised by both reviewers, I have a few additional points that I believe need to be addressed.

1. Detail on raw data: More detail is needed about the data used from the Woodland paper to allow readers to understand the nature of the variables being examined. Most importantly, what is the temporal resolution of the response variables? Are they snapshots, averages of multiple temporal replicates etc.? Give enough specifics to be able to understand the analysis without having to read the Woodland paper. Only the unimportant information should be left out, in that a reader should not have to read both papers together to understand this one.

2. Introduction: The second paragraph takes the reader down a road of what is the best measure of estuarine eutrophication, whereas this isn't the theme of the paper, it's what's the best driver of MA:TV. So why spend a third of the introduction leading the reader in this direction? This could be reduced down to a couple of sentences. This could then be replaced by currently-missing text that links the response variable with its predictors; i.e. information on the fact that several catchment factors could influence estuarine vegetation, and what these may be. This would then lead into the objectives paragraph (see also comments from reviewers on this).

Minor points:
Specify what cholorphyll a represents biologically at the outset.
L87: Specify 'S = 0'
L167-168: Very similar sentence to two sentences prior.
L298: Reference formatting.
Caption missing from Table 1.
Consistency US and UK english, e.g. 'ise' and 'ize'.

In light of the suggested revisions by the reviewers and myself, I have recommended 'major revisions'. I look forward to seeing a revised version of this manuscript.

Reviewer 1 ·

Basic reporting

The results are, in general, presented clearly; however, there is some confusion around the inclusion of chlorophyll a in their assessment. I suggest that the authors mention that (and explain why) they assessed chlorophyll a in their introduction (perhaps in final paragraph where the aim is outlined) and mention the analysis in the methods (which I’m assuming is the same as for their prediction of MA:TV).

Tables 1 and 2. Please reword your Table captions. It is insufficient just to say that it is a table of results. Ideally a table should be able to be read and understood without having to read the main body. Also expand on what you mean by pr(inc) and ß as many readers will not be readily familiar with your techniques and chosen notation. Please also indicate what the abbreviations you’re adopting (or write them in full) in your table, i.e., what does “Tf” represent? When you say “Values >0.75...” please be specific about what values you’re talking about, i.e., the Bayesian variable selection.

Table 3. Since you only measured 14 sites, it wouldn’t be too cumbersome, yet highly informative, if you could please show the export rates and land-use for all 14 sites rather than a summary.

Experimental design

The research question is generally well identified and methods are generally clear; however, as mentioned above, the authors need to also mention that they assessed chlorophyll in their introduction and methods.

Validity of the findings

No comment

Additional comments

General:

The authors present a study that proposes the ratio of macroalgae to vegetation (MA:TV) in South Australian estuaries as an indicator of ecological health and relate this to key catchment characteristics. A clear relationship between the % of fertilised land within a catchment and the MA:TV was identified. The manuscript is very well written and methods are appropriate for analysis. The potential factors influencing the MA:TV were selected apriori based on previous literature findings. Factors were examined for co-linearity, uninformative predictors removed and non-normal data transformed prior to analysis – all essential steps in having meaningful analysis. The results are, in general, presented clearly; however, there is some confusion around the inclusion of chlorophyll a in their assessment. I suggest that the authors mention that (and explain why) they assessed chlorophyll a in their introduction (perhaps in final paragraph where the aim is outlined) and mention the analysis in the methods (which I’m assuming is the same as for their prediction of MA:TV).


Specific:

Lines 46-48. Please cite the claims made in sentence beginning “Despite this, phytoplankton dynamics...”

Lines 59-61. Sentence beginning “The ratio of macroalgae...”. Please provide a short sentence explaining why the ratio of seagrasses and macroalgae is sensitive to nutrient loading.

Lines 65-69. Please clarify what is meant by ‘continental scale, stream-and-nested-catchment framework”. Perhaps re-word to say “catchment characteristics...”

Line 71. At this point you introduce assessing chlorophyll a data yet your aims don’t mention chlorophyll a? Please expand on your aims to mention the use of chlorophyll a (i.e., that you’re using it and why) in your introduction/aims.

Line 89. Please expand on the methodology used by ‘GUMLEAF’ to calculate loads and why this method is preferred to alternative methods.

Line 112. Jags is an acronym so change to JAGS.

Line 124. Where is the the R2 and p-value from? Is this also from heirarchical partitioning? If so then please explain the use of chlorophyll a data in your methods (and aims as mentioned above).

Line 145. This could be a good point to mention how, other than increasing nutrient loading, that fertilised land can affect MA:TV. Also total area of fertilised land doesn’t take into land use type and intensity or the attenuation capacity and lag time of the vadose zone (which will differ between catchments). Figure 2 from Elwan et al (2015) shows the spatial variability in attenuation between neighbouring sub-catchments. https://www.massey.ac.nz/~flrc/workshops/15/Manuscripts/Paper_Elwan_2015.pdf

Line 153. Whilst the dilution of the estuary may differ, won’t this occur of a gradient? So the areas closest to the mouth may still have high nutrient concentrations. Do you have estuary nutrient concentration data to support this claim?

Line 155. Are they ‘measures’ or does the sentence need re-wording? This sentence is currently confusing but seems crucial to understanding the following list. I’m currently not entirely sure why there is a list of attributes, is these hypotheses on factors which may be influencing the MA:TV ratio? The paragraph also seems incomplete without a brief mention on why each of these factors may affect MA:TV and the likeliness that they are impacting your systems.

Line 163. From Figure 1 it appears that the impact of increasing fertilised land ceases to change the MA:TV after ~20-40%, why might this be? Why doesn’t MA:TV change linearly with increasing fertilised land?

Line 172. That’s an interesting finding. I’d be curious to see what kind of vegetation composed the foreign forests you compared to (no need to comment on this in manuscript).

Line 187. Ecological consequences to increasing macroalgae i take it?

Line 222. Please explain this interaction model in your methods.

Line 229. Does MA:TV differ depending on where within the estuary it is surveyed? Or do your estimates cover the entire estuary?

Tables 1 and 2. Please reword your Table captions. It is insufficient just to say that it is a table of results. Ideally a table should be able to be read and understood without having to read the main body. Also expand on what you mean by pr(inc) and ß as many readers will not be readily familiar with your techniques and chosen notation. Please also indicate what the abbreviations you’re adopting (or write them in full) in your table, i.e., what does “Tf” represent? When you say “Values >0.75...” please be specific about what values you’re talking about, i.e., the Bayesian variable selection.

Table 3. Since you only measured 14 sites, it wouldn’t be too cumbersome, yet highly informative, if you could please show the export rates and land-use for all 14 sites rather than a summary.

·

Basic reporting

1. Hypotheses: I thought these could be better explained. Lines 65-69 has a dense and long sentence describing what the authors did, but there is no mention of what they expected to find. Also reiterating why this expectation might be useful from a management perspective would be helpful.

Experimental design

1. Data transformation: Please explain why the proportion data was not logit-transformed for hierarchical partitioning (see Warton, D. I. and Hui, F. K. C. 2011. The arcsine is asinine: the analysis of proportions in ecology. Ecology, 92: 3–10).
2. Statistical approaches: It would be useful if the authors described why the respective statistical methods (Bayesian variable selection and hierarchical partitioning) were used. These are appropriate, but it would be helpful for the readers to know what they do and why this is important for the data analysis of this study. For example, see Hatt et al. 2004: "This method allows identification of variables whose independent correlation with the dependent variable is strong, in contrast to variables that have little independent effect but have a high correlation with the dependent variable resulting from joint correlation with other independent variables." (Hatt et al. 2004. The influence of urban density and drainage infrastructure on the concentrations and loads of pollutants in small streams. Environmental Management, 34, 112–124.)
3. Data transformation: Please state which predictors were right-skewed, and thus log-transformed (Lines 104-105). You can refer to Tables 1-2 to save space (but see comment 1 above).
4. Site selection: It would be useful to quickly describe the selection criteria outlined in Woodland et al. (2015) here. This would help assuage any concerns that the sites had been “cherry-picked” for the non-linear response shown in Figure 1.

Validity of the findings

1. Validity of threshold response: The authors have identified a threshold of >20% of the catchment fertilized and estuarine macroalgal dominance (e.g. lines 122-123). While this seems reasonable, it would be good if this were tested objectively. Two approaches spring to mind: change-point analysis and non-linear regression ´curve´ fitting. For an example of how to use these two methods to validate an areal threshold see Burdon, F. J. et al. 2013. Habitat loss drives threshold response of benthic invertebrate communities to deposited sediment in agricultural streams. Ecological Applications, 23: 1036–1047. It should be noted that there are new ways to implement changepoint analyses in R (see Killick, R. & Eckley, I. A. 2004. Changepoint: an R package for changepoint analysis. Journal of Statistical Software 58, 1–19). Nonlinear regression fitting can make use of the "nls" and "AICc" functions in R. The latter function is in the "AICcmodavg" package. This would help describe objectively the shape of the relationship. It is important to note that the data should not be transformed for these analyses, because elucidating the actual threshold is the goal here (although OK to test a log-linear model with other candidate regression models). You can plot nonlinear regression lines a number of ways in R, one way uses "ggplot2".
2. Proximate and ultimate drivers of macroalgal cover: Although the authors discuss this issue (e.g. Lines 125-137), I wondered if a simple structural equation model (SEM) could not cast some more light on the interplay between catchment fertilization, river nitrogen loading, and estuarine macroalgae cover (keep in mind though caveats about site replication). For example, there should be indirect effects of catchment fertilization mediated through river nitrogen loading on estuarine macroalgae cover, but also a direct effect of catchment fertilization not related to river nitrogen loads (as hypothesized in Lines 156-161). SEM would be an objective way to assess causality, and its inclusion would add greater depth to the manuscript. Burdon et al. (2013) also make use of this statistical approach, but there are new R packages that can perform SEM analyses (e.g., see the "lavaan" package and the supplementary information in Grace, J. B., et al. 2014. Causal networks clarify productivity–richness interrelations, bivariate plots do not. Functional Ecology, 28: 787–798). This is important, because the “take home” message in Woodland et al. 2015 was that nitrogen load was the main driver of primary productivity in the same estuaries, but here you find that landuse (as an integrator of intensive agricultural activities) is a stronger predictor of estuarine macroalgal cover.
3. Residence times (Lines 213-214): How do the residence times of your systems compare to other estuaries around the world? It would be useful to quantify the "relatively" short duration here, and also help place your findings in the global context for the readers.
4. Mechanisms of threshold response: There is a rich literature on ecological stability and the abrupt transition from one system state to another. Although the authors do not have sufficient predictor variables or site replicates to test the underlying mechanisms of this complex system response objectively (but the simple SEM suggested above would be a move in the right direction), they could discuss their observation more with other related phenomena (e.g., state shifts to cyano-bacterial dominated algal assemblages in lakes). This would be helpful in further suggesting ways to move the science to a more predictive basis and away from description (which is useful as well - identification of thresholds have high utility for management). For a primer on this I can recommend Beisner et al. 2003. Alternative stable states in ecology. Frontiers in Ecology and the Environment 1(7): 376-382.

Additional comments

1. Lines 56-57. Is sedimentation and eutrophication of estuaries resulting from changes in catchment landuses not related to seagrass declines? Perhaps make this point a bit clearer.
2. Lines 65-69. This a long sentence. I would consider breaking it up and make the points clearer and more concise. Also see comment above under the "Basic Reporting".
3. Line 29. Delete "great".
3. Line 29. Add "...value by supporting..."
4. Line 30. Oxford comma between "...highly productive fisheries, and recreational...". Do you have a reference for these ecosystem services?
5. Line 31. "bear the brunt" is idiomatic. Consider rewording to "...estuaries are impacted by increased..."
6. Line 100. Replace "elided " with "discarded".
7: Lines 120-121. Reword to "was positively associated...".

---

## Round 0.2 · Minor Revisions

Many thanks for the improvements you have made to the manuscript from the previous reviews – it is much improved as a result. I sent the manuscript to one of the original reviewers (Dr. Burdon) for further comments, and he raised some additional points that require responses. I also noted a few very minor additional points below. As you'll see, these revisions are very minor. I look forward to seeing the revised version of your manuscript.

Minor points:

General notation: Please ensure consistent spacing of units (e.g. 0.5 m, not 0.5m).
L46: 'estuarine estuary'
L84: 'photographs'
L96: This example doesn't follow clearly from the 50% vaule given in the previous sentence. Please clarify.
L130: Note subscripted t.

·

Basic reporting

1. Hypotheses: The Authors have greatly improved the Introduction. Their Aims are stated unequivocally, and the utility of their approach to management clearly highlighted.

Experimental design

1. Data transformation: The Authors have explained that they did not logit-transform the proportion data because the “use of any transformation that achieves the unskewed-even spread empirically is sufficient”. There is a reasonable spread in the data which suggests this is probably true, but it would be more comforting if it were reported that the logit-transformation was attempted and did not qualitatively affect the results.
2. Statistical approaches: I have cited Ralph MacNally’s work on hierarchical partitioning, so am familiar with this contribution to the ecological literature. The example I provided previously comes from the R documentation authored by Walsh and MacNally, which I found useful when describing this method in my 2008 paper “The linkage between riparian predators and aquatic insects across a stream‐resource spectrum” (Freshwater Biology 53, 330-346). Nevertheless, it pleased me to find that the Authors have improved their descriptions of the analyses used in this manuscript. It is now clear why and how they have applied Bayesian variable selection and HP for this research.
3. Data transformation: OK, but see comments above re point 1.
4. Site selection: The Authors have provided the additional information required.

Validity of the findings

1. Validity of threshold response: The Authors have justified not objectively testing the stated threshold because “The key point here is that once catchment fertilization exceeds 20%, > 50% of the SAV is macroalgae (i.e., macroalgal dominated)”. Whilst I agree that the change in areal coverage is probably asymptotic, I would be much less confident to state that “this is just a phenomenological observation and nothing to do with a formal phase transition to another state per se”. There seems to be an element of confirmation bias in this view. Thresholds values are highly useful for management, and objective testing is not something to be dismissed out of hand. It has much less to do with “deepening our understanding”, and more with defining explicitly a value with utility for managers. I personally would find this work more useful if the threshold value were defined using changepoint analysis, as opposed to the current interpretation which is like ‘it seems to change somewhere around here’.
2. Proximate and ultimate drivers of macroalgal cover: The Authors have justified not using SEM here because of the low level of site replication. This is reasonable with data for only 14 sites. However, I would encourage them to consider using this approach in the future, given the high likelihood of complex interactions/contingencies with the stated predictor and response variables.
3. Residence times: Good stuff - references provided and a caveat issued about residence times.
4. Mechanisms of threshold response: The Authors have backed off from using “transition” in their article because it implies that the threshold elucidated is suggestive of a shift to a new ‘domain of attraction’, indicated by a much greater areal coverage of macroalgae (i.e., macroalgal dominance) in the studied estuaries. Whilst the Authors seem skeptical of such phenomena, it would seem irresponsible to discount this possibility out of hand. For one example, a stream study indicated that crossing a nutrient threshold may lead to synchronous declines in sensitive alga species with simultaneous increases in tolerant species associated with increasing enrichment (Taylor, J. M., King, R. S., Pease, A. A. and Winemiller, K. O. 2014. Nonlinear response of stream ecosystem structure to low-level phosphorus enrichment. Freshwater Biology, 59: 969–984).

Additional comments

The Authors have made many changes that have improved the manuscript. However, they have been resistant to certain suggestions. The changepoint analysis might show no clear change (or a large 95% CI) because of the relatively low site replication, but that would still be a result worth reporting, because validating the threshold would enable comparison with future studies.

---

## Round 0.3 · accepted · Accept

Many thanks for your revisions. I'm delighted to now accept your manuscript for publication. Rather than sending back to you for revisions, please note during the proofing stage to correct 'areal photographs' to 'aerial photographs' (L95). Also, please note the inconsistent use of 20% and 24% throughout the manuscript when referring to the changepoint. I think this is minor enough to leave for the proofing stage, but please choose one or the other for consistency.